# Enhancing LLM Reasoning via Vision-Augmented Prompting

**Ziyang Xiao**[1], **Dongxiang Zhang**[14],[*] **Xiongwei Han**[2]**, Xiaojin Fu**[2]**, Wing Yin Yu**[2]
**Tao Zhong**[2]**, Sai Wu**[14]**, Yuan Wang**[3]**, Jianwei Yin**[1]**, Gang Chen**[1]

[1] Zhejiang University    [2] Huawei Noah's Ark Lab
[3] School of Business, Singapore University of Social Sciences
[4] Hangzhou High-Tech Zone(Binjiang) Institute of Blockchain and Data Security
{xiaoziyang, zhangdongxiang, wusai, cg}@zju.edu.cn
{hanxiongwei, rocket.YuWingYin, zhongtao5}@huawei.com
fuxiaojin32@hotmail.com, Jessicawang36@gmail.com
zjuyjw@cs.zju.edu.cn

## Abstract

Verbal and visual-spatial information processing are two critical subsystems that activate different brain regions and often collaborate together for cognitive reasoning. Despite the rapid advancement of LLM-based reasoning, the mainstream frameworks, such as Chain-of-Thought (CoT) and its variants, primarily focus on the verbal dimension, resulting in limitations in tackling reasoning problems with visual and spatial clues. To bridge the gap, we propose a novel dual-modality reasoning framework called Vision-Augmented Prompting (VAP). Upon receiving a textual problem description, VAP automatically synthesizes an image from the visual and spatial clues by utilizing external drawing tools. Subsequently, VAP formulates a chain of thought in both modalities and iteratively refines the synthesized image. Finally, a conclusive reasoning scheme based on self-alignment is proposed for final result generation. Extensive experiments are conducted across four versatile tasks, including solving geometry problems, Sudoku, time series prediction, and travelling salesman problem. The results validate the superiority of VAP over existing LLMs-based reasoning frameworks.

## 1 Introduction

The human cognitive system is characterized by the presence of two specialized subsystems within the working memory: the phonological loop, which processes verbal information, and the visual-spatial sketchpad, which processes visual and spatial information [1]. Both of them play a crucial role in problem-solving by offering qualitatively distinct strategies for comprehending and manipulating information [2]. Recently, with the rapid advancement of LLMs, there have emerged various reasoning frameworks, such as Chain of Thought (CoT) [3], Self-consistent CoT (CoT-SC) [4], and Tree of Thoughts (ToT) [5]. Although these frameworks have shown impressive performance across a wide range of NLP tasks, they primarily focus on the verbal dimension with text-only representations, resulting in limitations in tasks that require visual and spatial interpretation (e.g., geometry problems or grid puzzles).

In this paper, we propose a novel dual-modality reasoning approach called **V**ision-**A**ugmented **P**rompting (VAP) that analogizes human cognition subsystems with the assistance of multimodal large language models (MLLMs). VAP takes textual problems as input and uses self-synthesized

---

[*]Corresponding author.

38th Conference on Neural Information Processing Systems (NeurIPS 2024).

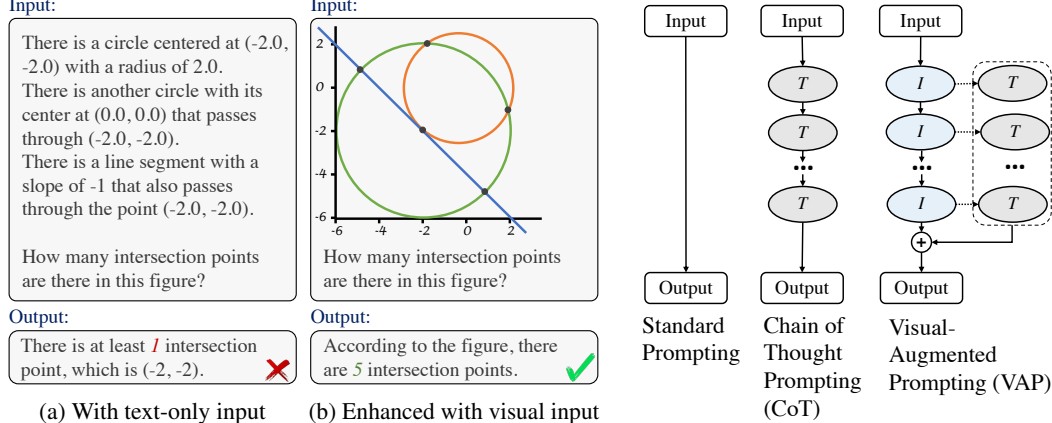

(a) With text-only input  (b) Enhanced with visual input

Figure 1: An example of solving the same Geometry Intersection Counting problem using different types of input. Outputs are derived from GPT-4; for brevity, only the conclusions are presented.

Figure 2: Compare with Standard Prompting and Chain of Thoughts Prompting(CoT).

images as an additional information channel to enhance reasoning. As shown in Figure 1, we employ the state-of-the-art GPT-4V(ision) [6] to solve a geometry problem. In this case, the model accurately deduces the correct answer when given both image and text inputs. In contrast, it generates an ambiguous answer with purely textual information. This example is analogous to the process of human cognition where it is a common practice to use images to enhance comprehension when handling geometry problems [7].

Our proposed vision-augmented prompting comprises three steps. Initially, LLMs generates a high-level plan for the following steps, including selecting an appropriate drawing toolkit and creating an initial image. To automate the procedure, we leverage the API documentation of external tools as context of LLM to facilitate drawing tool selection and figure synthesis. In the second step, VAP iteratively performs reasoning on the image, updates it, and generates an accompanying textual thought in each iteration. This process results in a chain of thoughts in both image and text modalities, as illustrated in Figure 2. Lastly, the final image and the chain of textual thoughts are jointly fed to the MLLM to derive a conclusive answer. To enhance robustness, we introduce a technique called self-alignment, where the MLLM describes the image content first, and the image channel is discarded if the self-description fails to align with the initial high-level plan.

We conduct extensive experiments across four versatile tasks, among which VAP establishes new state-of-the-art performance compared to other training-free LLMs-based methods. These four reasoning tasks include: (1) Geometry Intersection Counting in the domain of geometry word problems (+5.0% absolute accuracy gains); (2) Sudoku Puzzle as a logical reasoning task (+12.9%); (3) Time Series Prediction as a numerical analysis task (+9.9% in reducing mean absolute error); and (4) Travelling Salesman Problem as a classical NP-hard problem in the field of operations research (+1.8% in reducing the optimal gap).

## 2 Related Work

### 2.1 LLMs-based Reasoning

Improving the reasoning capabilities of LLMs such as GPT-4 [8], PaLM2 [9], and LLaMA2 [10] has been a hot topic in recent years. To achieve the goal, Chain-of-Thought (CoT) [3] breaks a reasoning task into a series of intermediate reasoning steps. Self-consistency [4] replaces the naive greedy decoding in CoT by sampling a diverse set of reasoning paths and selecting the most consistent answer. Tree of Thoughts (ToT) [5] and Graph of Thoughts (GoT ) [11] further allow LLMs to explore and combine thoughts in a structured manner. Cumulative Reasoning (CR) [12] decomposes a reasoning task into smaller components, which are solved in a cumulative and iterative manner.

Additionally, Chain-of-Experts [13] enhances reasoning capabilities through the collaboration of multiple LLMs.

## 2.2 Multimodal Large Language Models

The emergence of Multi-modal Large Language Models (MLLMs) such as GPT-4V(ison) [6] has fostered a new research landscape. Although the architecture of GPT-4V is not publicly available, substantial progress has been made in the open-source domain [14, 15, 16, 17]. These works usually fine-tune traditional text-based LLMs to align with other modalities. Similar approaches are also adopted by multimodal chatbots [18] and multimodal universal task solvers [15, 19] for vision-related tasks [20, 21, 22].

## 2.3 MLLMs-based Reasoning

According to a recent survey [23], reasoning based on MLLMs can be broadly categorized into LLM-aided visual reasoning and multi-modal Chain-of-Thought. LLM-aided visual reasoning focuses on solving traditional visual reasoning tasks with the assistance of LLMs. This includes tasks such as visual question answering [24], image segmentation [25], and video question answering [26]. Some works in this category, like ViperGPT [27], VisProg [28], and LLava-Plus [29], also use MLLMs to call external tools for enhanced reasoning, while their task settings differ from ours. On the other hand, the multi-modal Chain-of-Thought extends traditional prompting techniques to the multi-modal context. MM-ReAct [30] extends ReAct [31] to support multimodal data. The Visual Chain of Thought (VCoT) [32] uses CoT with vision-language grounding to bridge the gap in multi-step temporal reasoning. Compared with our work, VCoT is designed to bridge the logical gaps within sequential data and facilitate temporal reasoning in tasks like visual storytelling and WikiHow summarization.

## 3 Methodology

Mainstream LLMs-based reasoning frameworks, such as CoT, ToT, and GoT, only consider the verbal domain $\mathcal{L}$. To tackle more challenging problems with visual and spatial clues, we propose vision-augmented prompting (VAP) and extend the exploration space from a single domain $\mathcal{L}$ to a dual-modality domain $\mathcal{L} \cup \mathcal{G}$, where $\mathcal{G}$ represents the visual-spatial domain. It relies on the external image synthesis toolkit to automatically draw an image matching the text description of the input problem (Section 3.1). The core challenge lies in how to effectively navigate the joint space and maximize the success rate of problem solving, which will be presented in Section 3.2.

### 3.1 External Image Synthesis Toolkit

While visual-spatial information can be beneficial for reasoning as demonstrated in Figure 1, LLMs lack the inherent ability to visualize concepts. Therefore, it is necessary to leverage external image synthesis toolkit. Vision-augmented prompting incorporates graphic rendering tools that rely on logical programming to render images, including Python Turtle [33] for graphical visualizations, and Matplotlib [34] for drawing analytical figures. In addition to these third-party programmable tools, we further integrate image generative models, such as DALL·E 3 [35], to produce images directly from textual prompts. Consequently, we define the set of our drawing tools as $\mathcal{S}_T = \{\text{Turtle}, \text{Matplotlib}, \text{DALL·E 3}\}$. These tools will be utilized by our method through API calls.

### 3.2 Procedures of Vision-Augmented Prompting

We model the problem-solving process of VAP as an iterative reasoning process. With the aid of the image synthesis tools, VAP progressively updates the image according to the instruction provided by the language model. To maintain a coherent reasoning trajectory, a 'thought' is generated on each iteration. Subsequently, to derive the conclusive answer, the final image, the original problem, and the trajectory of iterative thoughts are sent to the MLLM to obtain the final answer. However, we find it challenging for LLMs to perform iterative multi-step drawing and reasoning directly due to the lack of a global view. To overcome this limitation, we introduce a planning step, where LLMs will create a high-level plan for subsequent steps. Furthermore, it is possible that the synthesized image could be

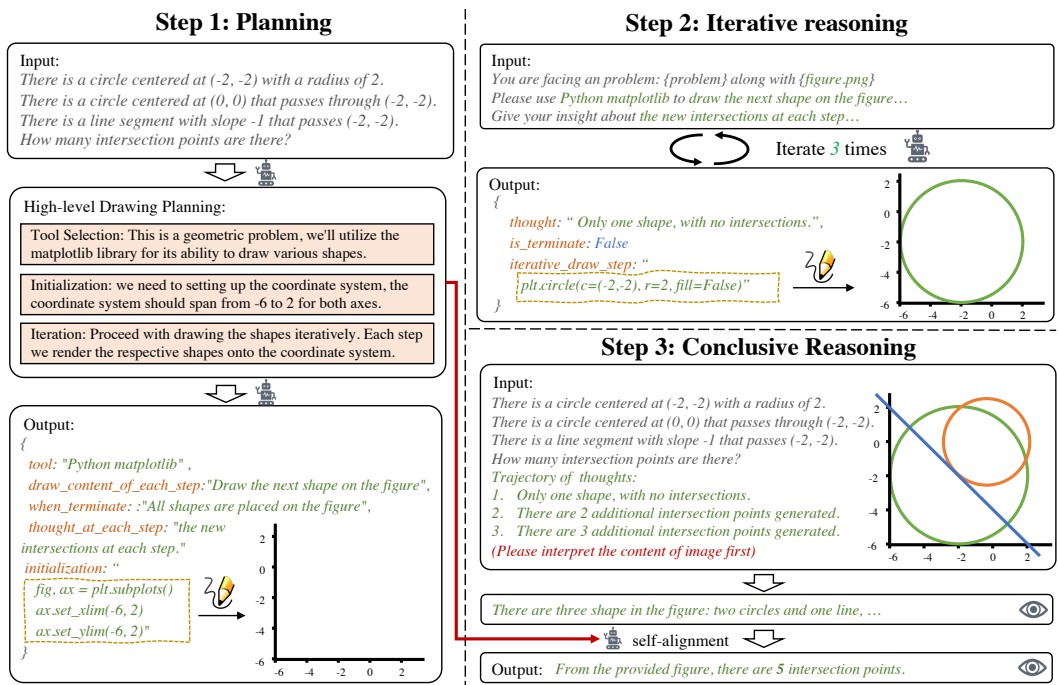

Figure 3: Illustration of the workflow in VAP.

disqualified and mislead the reasoning process. To alleviate the negative effect of such images, we introduce a technique named self-alignment to enforce MLLMs to literally describe the synthesized image and check whether its text description is aligned with the initial high-level plan. If disparity is detected, we discard the image and restart the reasoning process.

**Step 1: Planning** As demonstratrd in Figure 3, our method takes a textual problem $\mathcal{P}$ as input and firstly generates a high-level reasoning plan. In implementation, we first prompt the LLM to acquire a detailed description of the plan in natural language. The output of the plan consists of five key components: (1) an analysis of the problem's characteristics for visualization, along with the chosen drawing tool $T$ from the set of available tools $\mathcal{S}_T$; (2) A description of how to initialize the image using the chosen tool, accompanied by the corresponding API call instruction $I_0$; (3) the prompt $\mathbb{P}_d$ that drives the LLM to draw in each iterative step; (4) the prompt $\mathbb{P}_t$ that drives the LLM to think step by step during the iterative drawing process; (5) the termination condition $C$ for the iteration. To facilitate subsequent reasoning, we further transform the unstructured textual description of the planning into semi-structured JSON format. This step can be formalized as shown in Equation 1, where $\mathcal{F}_{planning}$ denotes the language model involved in this step with prompt engineering. With the instruction $I_0$, we can create the initial image $g_0 \in \mathcal{G}$ using the selected tool.

$$\{T, I_0, \mathbb{P}_d, \mathbb{P}_t, C\} \leftarrow \mathcal{F}_{planning}(\mathcal{P}, \mathcal{S}_T), \ g_0 \leftarrow T(I_0) \tag{1}$$

Note that both plan generation and JSON format transformation are implemented using LLMs with prompt engineering. Details of the prompt templates can be found in Appendix A.1.1.

**Step 2: Iterative reasoning** In the iterative reasoning step, the MLLM is specified by two prompts $\mathbb{P}_d$ and $\mathbb{P}_t$ generated in the previous step, along with the image synthesis tool $\mathcal{S}_T$ and the assigned termination condition $C$. We denote the MLLM in this step as $\mathcal{F}_{\{\mathbb{P}_d, \mathbb{P}_t, \mathcal{S}_T, C\}}$. In each iteration $t$, the MLLM takes the original problem $\mathcal{P}$, the partially-completed image $g_t$ and trajectory of thoughts $Z_t$ as input. The MLLM will then generate an instruction $I_t$ containing API calls used to update the image $g_t$. Regarding the update of the image, an accompanying 'thought' $z_t$ that provides a textual interpretation of the current state is generated. The process is formally depicted in Equation 2, with details of the prompt template presented in Appendix A.1.2. Here, $Z_t$ starts as an empty array and the new 'thought' $z_t$ is appended to the trajectory at each iteration.

$$\{z_t, I_t\} \leftarrow \mathcal{F}_{\{\mathbb{P}_d, \mathbb{P}_t, \mathcal{S}_T, C\}}(\mathcal{P}, g_t, Z_t), \ Z_{t+1} \leftarrow Z_t \cup \{z_t\}, \ g_{t+1} \leftarrow T(I_t) \tag{2}$$

We observe that the MLLM occasionally failed to follow the instructions provided in the prompt (e.g., repeatedly drawing the same shape), causing disruption to the entire iterative process. To address this issue and improve the stability of reasoning process, we enrich the context of MLLM by appending the input and output from the last step, which serve as illustrative examples for the MLLM to better follow the instructions.

**Step 3: Conclusive reasoning**   When the iterative reasoning terminates (i.e., condition $C$ is met), we proceed to the final step of conclusive reasoning using the synthesized image of step 2, denoted by $g_n$. Since we cannot guarantee $g_n$ is a perfect output and a disqualified $g_n$ may mislead the conclusive reasoning process, we introduce a technique named self-alignment to enforce MLLMs to literally describe the synthesized image and check whether its text description is aligned with the initial high-level plan. If disparity is detected, we discard the image and restart the reasoning process. If a qualified image cannot be obtained after a certain number of trials, VAP is degraded to simple input-output reasoning, without leveraging the visual channel.

As shown Equation 3, our conclusive reasoning takes $g_n$, the trajectory of thoughts $Z_n$, along with the original problem $\mathcal{P}$ as input and leverage the prompt template presented in Appendix A.1.3 to derive the final answer $\mathcal{A}$.

$$\mathcal{A} \leftarrow \mathcal{F}_{self-alignment}(\mathcal{P}, g_n, Z_n) \tag{3}$$

# 4   Experiments

We evaluate VAP using four diversified and challenging tasks, including Geometry Intersection Count, Sudoku Puzzle, Time Series Prediction, and Travelling Salesman Problem. We also provide a typical input-output example for each task in Appendix A.2.

These tasks share three LLM-based reasoning baselines, including standard prompting, chain-of-thought (CoT) prompting, and CoT with self-consistency (CoT-SC) prompting [4].

For CoT prompting, the reasoning process involves multiple intermediate steps. In the Geometry Intersection Counting task, we define each intermediate step as calculating the number of intersections between a pair of shapes. In the Sudoku Puzzle task, each step involves considering a position that violates the Sudoku rules. For Time Series Prediction and Travelling Salesman Problem, it is challenging to define specific step content. Therefore, we append a 'think step by step' instruction to the standard prompt and include an additional step to extract the solution from the generated thoughts.

Additionally, for CoT-SC prompting, when the task output is discrete (i.e., Geometry Intersection Counting, Sudoku Puzzle, and Travelling Salesman Problem), we sample $k$ answers and use the majority answer. When the task output is continuous (i.e., numeric value in the task of Time Series Prediction), we generate $k$ predictions and calculate their average as the final forecast value. The default $k$ is set to 10.

For fairness, we employ the 'GPT-4-vision-preview' as the underlying MLLM for these baselines and our VAP. The default temperature is set to 0. For methods that require sampling, such as SC, the temperature is set to 0.7.

## 4.1   Task 1: Geometry Intersection Counting

**Task Description**   The task determines the number of intersection points between a couple of geometric shapes described in natural language. For example, given the input "There is a line segment from $(-2.5, -1)$ to $(-0.5, -1)$ and a circle centered at $(-1.5, -1)$ with radius 1," the correct output should be "2", as there are 2 intersection points between the circle and the line segment.

**Task Setup**   We randomly sample 200 problem instances from Geometry Intersection Counting task in the BIG-bench benchmark[2] [36] . The performance metric used in this task is accuracy, which is defined as the percentage of problem instances in which the output number matches the ground truth.

---

[2] https://github.com/google/BIG-bench/tree/main/bigbench/benchmark_tasks/intersect_geometry

**Task-specific Baselines** We introduce two additional baselines specifically developed for solving geometry problems. The first is Inter-GPS [37], a symbolic reasoner [38] for geometry problems. The second is G-LLaVA [39], a MLLM specialized trained on geometry problems. Since geometry figures are not available in the problem input, G-LLaVA only leverages the textual modality for reasoning.

Table 1: Intersection Point Result.

| Method | Accuracy |
|---|---|
| Standard | 8.5% |
| CoT | 10.0% |
| CoT-SC$_{(k=5)}$ | 11.0% |
| CoT-SC$_{(k=10)}$ | 11.5% |
| CoT-SC$_{(k=20)}$ | 11.5% |
| Inter-GPS | 6.0% |
| G-LLaVA | 14.0% |
| **VAP** | **16.5%** |

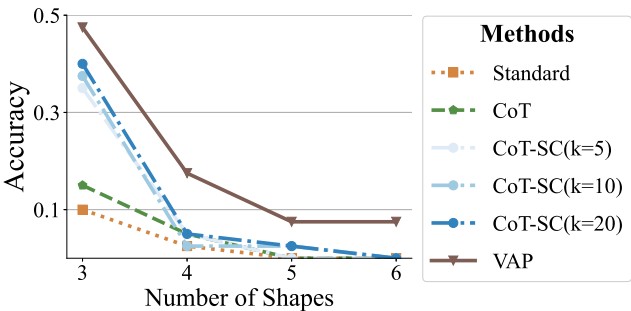

Figure 4: Accuracy with different number of shapes.

**Results** As shown in Table 1, Inter-GPS and G-LLaVA exhibit low accuracy, primarily because they are originally trained on dual-modality datasets and thus show limited effectiveness with text-only input. We also examine their performance when provided with an additional image input in Section 4.5, where the purpose is to validate the effectiveness of the synthesized images by our VAP algorithm. Among the general-purpose reasoning methods, standard prompting and CoT prompting yield relatively low accuracy (8.5% and 10.0% respectively). The CoT-SC method, even with a sample size of 20, only marginally improves accuracy by 1.5%. In contrast, VAP significantly outperforms these methods and achieves an accuracy of 16.5%. Furthermore, in the break-down analysis with increasing number of shapes, as illustrated in Figure 4, VAP demonstrates clear superiority over other LLM-based methods in more complex scenarios involving four or more shapes. We can see that the accuracy of all baselines is almost close to 0 in these complex scenarios.

### 4.2 Task 2: Sudoku Puzzle

**Task Description** In this task, an initial state of a Sudoku board is presented in natural language, with digits on filled cells and dots denoting empty cells. For example, in a $9 \times 9$ board, a row "..6 ......" only contains digit 6 in the third column. The reasoning process iteratively provides the information of the next action, in the form of "x y digit", where 'x' and 'y' specify the cell's coordinates on the board, and 'digit' is the number to be placed in that cell.

**Task Setup** We utilize the Sudoku puzzle generation program from the BIG-bench[3] to create a dataset that includes 150 Sudoku puzzles. For performance evaluation, we employ correct rate and collision rate as two metrics. The correct rate measures the accuracy in solving the puzzles, and collision rate assesses the frequency at which the position in given command is already filled with numbers, which indicates a violation of Sudoku rules.

**Task-specific Baselines** For a more comprehensive evaluation, we also incorporate Tree of Thought (ToT) [40] as an additional baseline. The implementation of ToT in this task is not troublesome because it is straightforward to decompose the thought process into a tree-structured representation for board games.

**Results** In Table 2, the standard prompting shows a low success rate of 18.0%, with a high rate of rule violations. This can be attributed to Sudoku's demands for reasoning and rule comprehension, which pose a challenge for LLMs relying solely on prompt engineering. The CoT and CoT-SC methods show improved performance, as they can, to certain extent, enhance the coherence of LLM reasoning by providing a reference of previous steps. The ToT algorithm is the most effective among these baselines, achieving a 22.6% success rate, owing to its tree-structured thought process. Notably,

---

[3] https://github.com/google/BIG-bench/tree/main/bigbench/benchmark_tasks/sudoku

VAP outperforms all other methods with a correct rate of 35.5%. VAP also reduces the collision rate, implying that integrating the image modality is helpful for LLMs to understand game rules.

Table 2: Sudoku Result.

| Method | Correct rate↑ | Collision rate↓ |
|---|---|---|
| Standard[†] | 18.0% | 68.7% |
| CoT | 17.3% | 64.7% |
| CoT-SC$_{(k=5)}$ | 20.6% | 48.7% |
| CoT-SC$_{(k=10)}$ | 20.6% | 47.3% |
| ToT | 22.6% | 44.0% |
| VAP | **35.5%** | **26.0%** |

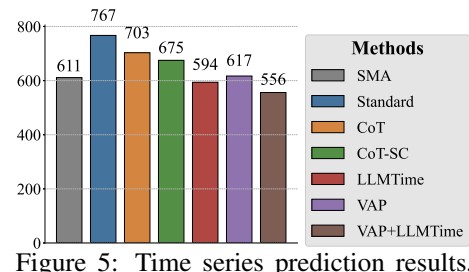

Figure 5: Time series prediction results, with y-axis indicating MAE.

### 4.3 Task 3: Time Series Prediction

**Task Description**  To evaluate the numerical analysis capabilities, we regard the LLM as a time series predictor. In this task, a sequence of data points is provided, and the objective for the LLM is to predict the next $n$ values in the series.

**Task Setup**  The dataset for this task is sourced from the Darts library [41], which includes a curated collection of 8 real univariate time series datasets. We configure the window size to 100 data points and predict the subsequent $n = 8$ values in the series. The performance is evaluated using the Mean Absolute Error (MAE) metric.

**Task-specific Baselines**  We chose the LLMTime algorithm [42] as the most cutting-edge method for comparison, where a data rescaling strategy and a tokenization trick is introduced to enhance numerical precision. For instance, the sequence "8.05, 1, 35" will be tokenized as "8 0 5 , 1 0 0 , 3 5 0 0". We also adopt the simple moving average (SMA) as a traditional statistical baseline for time series forecasting.

**Results**  As shown in Figure 5, SMA is recognized as a traditionally effective method for time series prediction, surpassing most LLM-based methods with an MAE of 611. Among the LLM-based approaches, compared to standard prompting, CoT-SC approach significantly reduces the MAE from 767 to 675, justifying the effectiveness of integrating auto-regression and sample strategy. It is noteworthy that LLMTime, as a preprocessing technique for LLMs to handle sequential data, performs effectively. Its performance even marginally exceeds that of SMA. In the result, VAP outperforms the majority of LLM-based methods and is slightly inferior to LLMTime. However, considering that VAP is orthogonal to LLMTime, we introduce a new method that combines the same preprocessing technique with VAP. The results show that this combined version, VAP+LLMTime, outperforms all comparison algorithms.

### 4.4 Task 4: Travelling Salesman Problem

Next, we test our method on travelling salesman problem (TSP), an NP-hard combinatorial optimization problem that poses challenges for neural network solvers [43]. The input for TSP instances consists of a set of coordinates, and the output is a sequence of city indices representing the route for the salesman.

**Task Setup**  We use Euclidean TSP instances with 10 and 20 cities as our testset. For each city size, 100 instances are generated by a program, with coordinates uniformly distributed within a $[0, 100]^2$ integer grid. The average path length and the gap from the optimal solution are used as metrics.

**Task-specific Baselines**  We introduce four traditional TSP solvers as our task-specific baselines: Gurobi, an exact TSP solver using mixed integer programming; Nearest neighbour (NN) algorithm, which greedily selects the closest city; Fastest insertion (FI), an efficient insertion method; and a Random baseline that chooses paths randomly for performance benchmarking. Here, note that the

output of LLM-based methods (i.e., Standard, CoT, CoT-SC, and VAP) might violate constraints of TSP (e.g., generating duplicate cities). In such cases, we randomly select an unvisited city to continue the solution.

Table 3: TSP task performance.

| Method | | Traditional | | | | LLMs-based | | | |
|---|---|---|---|---|---|---|---|---|---|
| | | Gurobi | Random | NN | FI | Standard | CoT | CoT-SC | VAP |
| N=10 | Length | **294.8** | 529.6 | 327.4 | 306.8 | 327.5 | 327.2 | 325.0 | **312.4** |
| | Gap | **0.0**% | 79.7% | 11.1% | 4.1% | 11.1% | 11.0% | 10.4% | **6.0**% |
| N=20 | Length | **383.7** | 989.5 | 448.2 | 424.2 | 544.5 | 547.2 | 541.9 | **499.3** |
| | Gap | **0.0**% | 170.7% | 16.8% | 10.6% | 41.9% | 42.6% | 41.2% | **30.2**% |

**Results**    As shown in Table 3, the performance of standard prompting is comparable to the Nearest Neighbour algorithm when applied to TSP with 10 cities. CoT demonstrates very slight improvement over standard prompting. This might be attributed to the TSP's requirement for a heuristic and complex thought process, which is challenging for CoT to replicate. VAP outperforms all LLM-based algorithms, achieving a 6.0% gap for optimality, close to the Fastest Insertion's performance. It's worth noting that when the number of cities in a problem instance increases to 20, the performance of LLM-based algorithms significantly decreases as the search space of TSP grows exponentially with the problem size. Such a huge search space poses a significant challenge for LLMs not specifically tailored to solve TSP. Nonetheless, VAP's superiority over other LLM-based methods is still evident when the problem size increases.

### 4.5 Effectiveness of the Synthesized Images

To demonstrate the quality of synthesized images, we first present several illustrative examples in Figure 6 and then provide numerical experiments in the following.

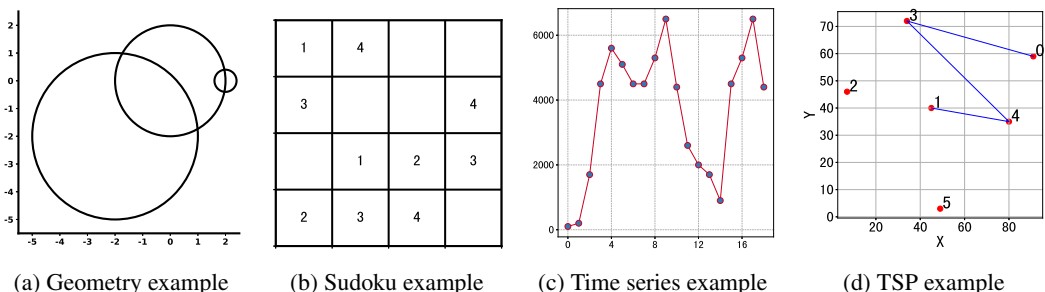

(a) Geometry example          (b) Sudoku example          (c) Time series example          (d) TSP example

Figure 6: Examples of images synthesized by our VAP among the four tasks.

In the first experiment to evaluate the quality of synthesized images, we use the rate of integrity as the metric, which refers to whether the generated image contains all the elements described in the original problem. This is a relatively objective metric for human annotators to reach consensus. As shown in Table 4, the images synthesized by VAP demonstrate a high integrity rate across various tasks. Notably, for Time Series Prediction and TSP, the module achieves an impressive integrity rate of 100% and 98.0%, respectively. We also evaluate the performance improvement of VAP when provided with the ground truth image. As shown in Table 4, the column of 'With ground truth image' refers to the performance boost rate. If the drawing errors had been corrected, we can observe notable potential for improvement. These results underscore the crucial role of accurate image rendering in enhancing the final output quality of the VAP.

In the next experiment, we provide the synthesized images by our VAP as additional input for the geometry solvers Inter-GPS and G-LLaVA involved in task of Geometry Intersection Counting. As shown in Table 5, these two approaches can greatly benefit from our synthesized images. Their remarkable accuracy boosting validates the effectiveness of the images generated by VAP.

Table 4: Performance of Drawing Module.

| Task | Integrity | With ground truth image |
|------|-----------|-------------------------|
| Geometry | 83.5% | 5.8% (17.0% → 18.0%) |
| Sudoku | 89.3% | 5.1% (35.5% → 37.3%) |
| Time Series | 100.0% | - |
| TSP | 98.0% | 1.3% (30.2% → 30.6%) |

Table 5: Performance of dedicated geometry solvers with (denoted by $^\dagger$) and without synthesized images.

| Method | Accuracy | Accuracy$^\dagger$ |
|--------|----------|--------------------|
| Inter-GPS | 6.0% | **17.0%** |
| G-LLaVA | 14.0% | **20.5%** |

## 5 Ablation Study

In the ablation study, we evaluate the effectiveness of three steps, including high-level planning, iterative reasoning and self-alignment in conclusive reasoning. Note that removing the planning step refers to prompting the LLM to draw the image step-by-step without providing a plan in advance. Removing iterative reasoning refers to directly prompting the LLM to generate an entire image and then proceeding to the conclusive reasoning step without intermediate results.

As shown in Table 6, we report the core metric for four tasks: accuracy for Geometry Intersection Problems, correct rate for Sudoku puzzles, MSE for Time Series Prediction, and average tour length for the TSP with 10 cities. The results demonstrate that removing any of the key components causes a performance degradation, except for self-alignment in Time Series Prediction. This is because there is no error in the drawing module, as discussed in Subsection 4.5. Hence, in this scenario, self-alignment plays no effect. Interestingly, removing planning step causes the most significant performance drop for Geometry Intersection Problems and TSP. The reason is that planning is crucial for visualizing problems requiring precise spatial relationships. Furthermore, iterative reasoning proves to be an essential component, as its removal leads to considerable performance degradation across all tasks. This finding highlights the importance of our view that all problem-solving is a step-by-step process. Self-alignment plays an important role in the task of Sudoku. For example, the LLM will occasionally draw the incorrect coordinate range for the Sudoku board, resulting in a board with the wrong size. However, with self-alignment, the model first describes the board size in the figure and cross-checks this against the original plan, preventing such issues.

Table 6: Ablation study results. Each column represents a task and the cell indicates the performance when removing a module in VAP.

| | Geometry | Sudoku | Time Series | TSP$_{(N=10)}$ |
|------|----------|--------|-------------|----------------|
| VAP | 16.5% | 35.5% | 556 | 312.4 |
| w/o planning | 12.0% | 24.7% | 578 | 322.0 |
| w/o iterative | 14.5% | 19.9% | 582 | 321.3 |
| w/o self-alignment | 16.0% | 25.1% | 556 | 315.9 |

## 6 Limitations of VAP

Even though VAP has been shown to outperform other LLM-based approaches in a set of versatile tasks, there still exists a noticeable performance gap between VAP and task-specific approaches. However, these tailored approaches require nontrivial efforts for domain customization and is not able to support other types of reasoning tasks. In contrast, VAP is a lightweight and training-free framework. With negligible customization cost, VAP is general enough to handle a spectrum of complex reasoning tasks with visual and spatial clues.

The other limitation is the black-box functionality of VAP, even though the framework is inspired by the cooperation of two subsystems in different regions of human brain that cooperate with each other. In Appendix A.3, we provide two case studies as examples to provide certain insights and justify the effectiveness of VAP. It would be an interesting future research direction to explore the interpretability of VAP and its relationship with cognitive science.

# 7 Conclusion

In this paper, we introduced a novel reasoning framework called Vision-Augmented Prompting (VAP), designed to enhance the reasoning capabilities of large language models (LLMs) by emulating the human cognitive system's dual modality processing. VAP leverages the external image synthesis tools to generate visual representations that augment the textual input. The procedure of VAP follows a three-step algorithm: first, it automatically generates a high-level plan for reasoning; second, it engages in iterative drawing and reasoning based on the partially-completed image; and finally, original problem, all thoughts and generated image are jointly to derive a solution. To enhance robustness, we introduced a self-alignment technique, where the MLLM describes the image content, and the image channel is discarded if the self-description fails to align with the initial high-level plan. We conducted extensive experiments across four diverse reasoning tasks: intersection counting in Geometry Intersection Problems, Sudoku Puzzles, Time Series Prediction, and the Travelling Salesman Problem. The results demonstrated the effectiveness of VAP, establishing new state-of-the-art performance compared to other training-free LLM-based methods.

## Acknowledgments and Disclosure of Funding

This work is supported by the National Key Research and Development Project of China (2022YFF0902000) and the Fundamental Research Funds for the Central Universities (226-2024-00145 and 226-2024-00216).

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

# A  Appendix

## A.1  Prompt Engineering of VAP

### A.1.1  Planning

The planning step's prompt is divided into two sub-steps. First, a high-level plan's textual representation is generated using the following prompt. Here, the '{examples}' and '{problem}' are placeholders for the few-shot examples (which will be discussed later) and the problem input, respectively.

Your role is to visualize a problem by creating an image that represents the problem description accurately using a specific tool. This task involves drawing an image that encapsulates all the details provided in the problem description.

The drawing will be executed through an iterative process. This means you will develop the image in a step-by-step manner, ensuring that each element of the problem is represented accurately.

Before beginning the drawing, you are required to outline a high-level plan for how you will approach the drawing process. This plan should cover three essential aspects:

1. Tool Selection - The choice of software or tool you will use for drawing. Here are drawing tools available:

- Python Matplotlib: a powerful Python library used for creating a wide variety of static, animated, and interactive visualizations. It is widely utilized in data science and engineering for generating high-quality plots and graphs.

---

//=== Continued from previous page ===
- Turtle: a Python library that provides a simple way to draw graphics and shapes using a virtual "turtle" that moves around the screen. Inspired by the Logo programming language, it is an excellent tool for teaching programming concepts through visual feedback.
- DALLE3: an advanced image generation model, capable of creating detailed and imaginative images from textual descriptions. It leverages deep learning to understand and produce highly realistic or fantastical scenes based on user prompts.

2. Initialization Approach: How you will begin your drawing, focusing on setting up the initial state of the image.

3. Iterative Drawing Approach: A detailed description of how you will iteratively add details to the image step by step.

Output Format: Your plan should be organized into three distinct paragraphs, each starting with the respective headings: Tool selection:, Initialization:, and Iteration:.

{examples}

Problem Description: {problem}

---

The few-shot examples are optional to improve the LLM's ability to handle various tasks. The planning step plays a crucial role in the overall reasoning process, therefor, we use a one-shot example as our default setting (the same as other baselines for fairness). For the planning step, we use several pairs of problem input and textual high-level plan as examples. For other standard prompting, we use the problem input and answer output as examples. For CoT prompting and CoT-SC prompting, we use problem input, trajectory of thoughts, and answer output as examples.

Subsequently, a semi-structured plan in JSON format is extracted using the following prompt. Here, {drawing_plan} is the placeholder of the high-level plan generated before.

---

You are tasked with visualizing problems by creating images. Each image should be constructed step by step, based on a detailed plan you've previously prepared.

You should first review your plan, and start by examining the high-level plan you've made for the iterative drawing process.

Your plan is as following: {drawing_plan}.

Your task is to extract meta-information from your drawing plan and format it as a JSON string.

Follow this strict JSON structure in your output:

{
"tool": "Name of the tool you're using",
"draw_content_of_each_step": "Describe what you'll draw in each step",
"when_terminate": "Criteria for completing the drawing",
"thought_at_each_step": "Your thought process at each step",
"initialization": "Initial setup before starting the drawing"
}

PLEASE ENSURE YOUR JSON OUTPUT IS CORRECTLY FORMATTED! Avoid including extraneous words or characters outside the JSON structure.

### A.1.2 Iterative reasoning

The prompt for the iterative reasoning step is as follows. Please note that current cutting-edge MLLMs cannot embed images at arbitrary positions of prompt. Therefore, we provide the image in context and refer to it in prompt description accordingly.

> You are a problem visualizer, tasked with drawing an image based on a problem description using a specified tool.
> You will draw the image accurately, ensuring that all information in the problem description is included in the image. This process will be done iteratively. Each iteration step will involve drawing a specific content of the figure using the tool.
> The problem description is as follow: {problem}
> Here is the partially completed figure given in context. {figure.png}. Please refer to it during your thought process and image updates.

> //=== Continued from previous page ===
> For each step, provide your thoughts in {thought_at_each_step}.
> Next, update the image content {draw_content_of_each_step} using API calls with the tool tool.
> Output Format:
> Your output should be a JSON string. Strictly follow the JSON format provided below.
> {
> "thought": "Your thought on this iteration",
> "is_terminal": false, // true if this is the last iteration, otherwise false. The terminition condition is {when_terminate}
> "iterative_draw_step": "Description of what was drawn or modified in this step"
> }
> PLEASE ENSURE YOUR JSON OUTPUT IS CORRECTLY FORMATTED! Do not include any extra words or comments.

### A.1.3 Conclusive reasoning

First, we apply self-alignment by prompting the MLLM to describe the content of the image.

> Please describe the details of the given image accurately and comprehensively. Include all visible objects and elements without omitting any details, and avoid adding any imaginary or non-existent objects. Ensure that the description is as detailed as possible, faithfully capturing the essence of the image provided.
> Here is the given image {image.png}.
> Your description is as follow:

Next, the MLLM is prompted to function as a binary classifier and is tasked with checking whether the description aligns with the initial high-level plan. The prompt is shown as follow.

> You are an artist who visualizes problems.
> You have made a plan for your drawing: {drawing_plan}
> The final content you drew is described as: {self_description}
> Now, you should determine whether the description aligns with your initial drawing plan. Pay close attention to the details in both.
> Your output should be a single word: 'true' if they align, and 'false' if they do not. Do not provide any additional comments or explanations.
> Here is your answer:

The prompt of conclusive reasoning step is as follow.

> Here is the problem: {problem}
> Its visualization is given in context. {figure.png}
> The description of the image is as follows: {self_description}
> The trajectory of thoughts is as follows: {thoughts_trajectory}
> Provide your final answer:

### A.1.4 Prompt used in ablation study

In the ablation study, we investigate the impact of the planning step by removing it from the process. The removal of the planning step results in the loss of access to meta-information, such as the selected tool, which can no longer be incorporated into the iterative reasoning prompt template. To address this issue, we employ an alternative prompt template specifically designed for iterative reasoning in the absence of the planning step. This alternative template allows for the continuation of the iterative reasoning process without relying on the meta-information that would have been provided by the planning step. The specific prompt we used is shown as follow.

> You are a problem visualizer, tasked with drawing an image based on a problem description using a specified tool.
> You will draw the image accurately, ensuring that all information in the problem description is included in the image. This process will be done iteratively. Each iteration step will involve drawing a specific content of the figure using the tool.
> The problem description is as follow: {problem}
> Here is the partially completed figure given in context. {figure.png}. Please refer to it during your thought process and image updates.
> For each step, provide your thoughts according to this problem.
> Next, update the image content according to this problem using Python API calls.
> Output Format:
> Your output should be a JSON string. Strictly follow the JSON format provided below.
> {
> "thought": "Your thought on this iteration",
> "is_terminal": false, // true if this is the last iteration, otherwise false
> "iterative_draw_step": "Description of what was drawn or modified in this step"
> }
> PLEASE ENSURE YOUR JSON OUTPUT IS CORRECTLY FORMATTED! Do not include any extra words or comments.

### A.2 Examples of Four Experimental Tasks

This section presents the input-output examples for the four tasks involved in the experiments.

#### A.2.1 Geometry Intersection Counting

**Input:** *There is a circle centered at (-1.5, -1.0) with radius 3.0. There is a polygon with coordinates [(-2.2, 4.0), (-3.2, -0.4), (2.4, -2.6), (3.8, 4.1)]. There is a line segment from (0.9, 3.3) to (-1.3, 3.4). How many intersection points are there?*

**Output:** *2*

#### A.2.2 Sudoko Puzzle

**Input:** *Sudoku puzzle Fill the dots with digits "[1-4]". Digits cannot repeat in the same row, column or 2x2cell. Each cell is separated by spaces or empty lines. Specify the new digits with a command: "<x> <y> <digit>", removing the quotes. The top left corner has coordinates "1 1". For example the command to add the digit "4" to the bottom left corner is "1 4 4".*

*Please pay attention to the information given in the image, especially the positions of the X-axis and Y-axis.*

*Board:*

*24 31*

*.3 ..*

*.1 2.*

*.2 1.*

*Command:*

**Output:** *1 1 4*

### A.2.3 Time Series Prediction

**Input:** *You are a time predictor. The user will provide a sequence and you will predict the next value(only one value needed). The sequence is represented by decimal strings separated by commas.*

*Please continue the following sequence without producing any additional text. Do not say anything like 'the next value in the sequence are', just return the numbers. Sequence:*

*458.0,387.0,427.0,565.0,465.0,445.0,450.0,556.0,500.0,452.0,435.0,554.0,510.0,433.0,453.0,548.0,*

*486.0,453.0,457.0,566.0,515.0,464.0,431.0,588.0,503.0,443.0,448.0,555.0,513.0,427.0,473.0,526.0,*

*548.0,440.0,469.0,575.0,493.0,433.0,480.0,576.0,475.0,405.0,435.0,535.0,453.0,430.0,417.0,552.0,*

*464.0,417.0,423.0,554.0,459.0,428.0,429.0,534.0,481.0,416.0,440.0,538.0,474.0,440.0,447.0,598.0,*

*467.0,439.0,446.0,567.0,485.0,441.0,429.0,599.0,464.0,424.0,436.0,574.0,443.0,410.0,420.0,532.0,*

*433.0,421.0,410.0,512.0,449.0,381.0,423.0,531.0,426.0,408.0,416.0,520.0,409.0,398.0,398.0,507.0,*

*432.0,398.0,406.0,526.0,*

**Output:** *438.0*

### A.2.4 Travelling Salesman Problem

**Input:** *You are an TSP solver. Solve a TSP instance:*

*Here are 10 point in a city, each point is represented as a pair of numbers, indicating its coordinates in the specified space. The coordinations are as following:*

*0: (0.9371, 0.1482)*

*1: (0.6345, 0.2510)*

*2: (0.2628, 0.7120)*

*3: (0.1545, 0.9067)*

*4: (0.2827, 0.8081)*

*5: (0.9533, 0.9086)*

*6: (0.4199, 0.7617)*

*7: (0.6315, 0.0414)*

*8: (0.8694, 0.5878)*

*9: (0.7709, 0.7211)*

*What is the shortest possible route that visits each city exactly once and returns to the origin city?*

*Your answer should be the permutation of all city index (separated by comma). The start point is 0. Please give your answer directly, don't use any other tools.*

**Output:** *0,7,1,8,9,5,2,6,4,3*

## A.3 Interpretability of VAP

To better understand how VAP facilitates problem-solving, we conduct two experimental studies to gain a deeper insight.

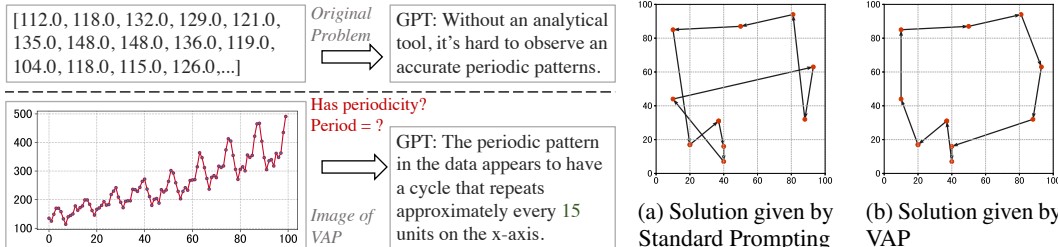

Figure 7: Detect periodicity with and without VAP.

(a) Solution given by Standard Prompting

(b) Solution given by VAP

Figure 8: A case study of TSP task.

Firstly, we focus on Time Series Prediction task. Figure 7 shows the Air Passengers Dataset from Darts. On the left, we have the textual representation of the sequence alongside an image drawn by VAP. We ask GPT-4V to model the periodicity of this sequence, which is an important factor in Time Series Prediction. Despite the clear periodic pattern in this sequence, GPT-4V struggles to recognize this periodicity when provided only with the textual problem. In contrast, when presented with the image, GPT-4V could easily identify the sequence's periodicity. This study demonstrates how VAP can provide a more efficient and clear information format, thereby enhancing the model's reasoning ability on prediction tasks.

The second study focuses on the TSP, where we visualize two solutions for the same TSP instance in Figure 8. One solution is generated using standard prompting, while the other is given by VAP. The comparison of the two solutions reveals that the standard prompting results in numerous crossing paths, which is proven to be suboptimal in Euclidean TSP [44]. In contrast, the VAP solution has few crossing paths. This improvement is likely due to the clear visualization of visited partial paths, which provides the model with heuristic cues to easily exclude bad choices.

## A.4 More Experimental Results

### A.4.1 Effiency of VAP

Rregarding the computational efficiency of VAP, we conduct an experiments comparing the time consumption and accuracy of various methods across geometry and Sudoku tasks. Table 7 presents the results of this analysis.

Table 7: Efficiency and Accuracy Comparison

| Method | Geometry | | Sudoku | |
|---|---|---|---|---|
| | Time Usage | Accuracy | Time Usage | Correct Rate |
| Standard | 0.2 s | 8.5% | 0.3 s | 18.0% |
| CoT | 0.5 s | 10.0% | 0.8 s | 17.3% |
| CoT-SC (k=5) | 2.3 s | 11.0% | 4.1 s | 20.6% |
| CoT-SC (k=10) | 4.5 s | 11.5% | 8.8 s | 20.6% |
| ToT (n_children=5) | - | - | 9.0 s | 22.6% |
| VAP | 4.1 s | **16.5%** | 9.5 s | **35.5%** |

Although VAP is computationally intensive, its time usage is comparable to ToT and CoT-SC (k=10) in the Sudoku task. Interestingly, for the simpler geometry task, VAP demonstrates faster performance than CoT-SC (k=10). This efficiency can be attributed to VAP's unique structure: despite its large context (including tool instructions, thought trajectory, and image encoding), the output is concise, consisting primarily of API calls and immediate thoughts. This leads to rapid inference during each step, as we observed that time usage is predominantly influenced by the number of decoded tokens. Given VAP's superior effectiveness on these tasks, we posit that the observed performance gap in computational efficiency is acceptable.

### A.4.2 Different Foundational Models

Our experimental setup employs a unified VLLM. However, we observed that VAP necessitates a MLLM to process visual-text input, whereas other baselines solely require textual input. Considering that traditional LLMs are expected to outperform MLLMs in text-only tasks, we introduced GPT-4 and LLaMA 3 8B as additional LLMs to ensure a more comprehensive and equitable comparison. Table 8 presents the results of this expanded analysis on geometry intersection counting task.

Table 8: Performance on Geometry Task over Different Foundational Models

| Method | Accuracy (GPT-4v) | Accuracy (GPT-4) | Accuracy (LLaMA 3) |
|---|---|---|---|
| Standard | 8.5% | 10.0% | 7.0% |
| CoT | 10.0% | 11.0% | 8.0% |
| CoT-SC (k = 5) | 11.0% | 11.5% | 8.0% |
| CoT-SC (k = 10) | 11.5% | 11.5% | 8.0% |
| CoT-SC (k = 20) | 11.5% | 11.5% | 8.0% |
| VAP | **16.5%** | - | - |

The findings reveal several noteworthy points. First, GPT-4 demonstrated a marginal improvement in baseline performance compared to GPT-4v, albeit still significantly lower than VAP. Conversely, LLaMA 3 8B exhibited reduced accuracy, which can be attributed to its smaller model size potentially limiting its generalization capabilities for this particular reasoning task. While larger variants of LLaMA 3 might yield superior results, current hardware constraints precluded their evaluation. Interestingly, simpler methods such as standard prompting show greater sensitivity to model variations compared to more complex approaches like CoT with CoT-SC. Despite these differences, we believe that the fundamental conclusions of our study remain valid.

