# OpenReview forum: "Enhancing LLM Reasoning via Vision-Augmented Prompting"
_NeurIPS.cc/2024/Conference — NeurIPS 2024 spotlight_

### Official Review · Reviewer_gmDj · 2024-07-04

**Soundness:** 2
**Presentation:** 3
**Contribution:** 3
**Rating:** 4
**Confidence:** 4

**Summary:**

Traditional large language models (LLMs) struggle with tasks requiring visual and spatial interpretation based solely on text. This study introduces a visual-augmented prompting (VAP) strategy, using an external image generation tool to iteratively create intermediate visual representations that aid reasoning. VAP's effectiveness is validated in four tasks: (1) Geometry Intersection Counting, (2) Sudoku Puzzles, (3) Time Series Prediction, and (4) the Traveling Salesman Problem.

**Strengths:**

(1) The VAP method is simple yet effective and well-motivated, aligning with human cognitive processes. It promisingly integrates both intermediate textual and visual results to enhance the accuracy and interpretability of the reasoning process.

(2) Results clearly show that VAP significantly outperforms all baselines across four tasks.

(3) The manuscript is well-written and easy to follow.

**Weaknesses:**

(1) Efficiency of Iterative Reasoning: The author does not mention the time consumption of the iterative reasoning process of VAP. Additionally, I am curious about the trade-off between the number of images that need to be drawn and model performance.

(2) More LLM models (such as LLaMA 3, GPT4) need to be incorporated for more comprehensive comparison.

(3) Effect of Different Figure Drawing Tools: I am interested in understanding how different graphic rendering tools for image drawing affect the final reasoning results.

(4) Method Scalability on Complex Geometry Problems: Figure 4 shows a significant accuracy drop for VAP when the number of shapes exceeds three. This raises concerns about the scalability of VAP in handling complex geometry problems.

(5) Although VAP performs better than the baselines, its overall performance is still poor and has a noticeable gap compared to traditional methods.

**Questions:**

For weakness (2), the author claimed that using GPT-4V as the foundational model for the baseline methods ensures fairness. However, VAP requires an MLLM to handle visual-text input, whereas all other baselines only require textual input. I believe traditional LLMs should outperform MLLMs for text-only tasks. Therefore, the author is expected to incorporate a wider range of LLM models (such as LLaMA 3, GPT4) for comparison.

I am curious about the potential of VAP in solving real-world reasoning problems, where the model needs to generate more photorealistic images (not just diagrams) to benefit reasoning.

**Limitations:**

The author has a section that addresses most of the limitations of the VAP method. However, I suggest that if the VAP method is time-consuming in terms of the iterative figure drawing process, the author should discuss the efficiency problem in the limitations section.

---

> ### Author Rebuttal · Authors · 2024-08-06
>
> We sincerely appreciate the constructive comments on our work, which have helped us to enhance the paper. The following are our detailed responses to each comments.
>
> ### W1.1:Efficiency of iterative reasoning
>
> This is a good point. We added an efficiency experiment for VAP in our revision. The results for the geometry and Sudoku tasks are shown below:
>
> |               | Geometry |          | Sudoku |              |
> | ------------- | -------- | -------- | ------ | ------------ |
> |               | Time     | Accuracy | Time   | Correct rate |
> | Standard      | 0.2 s    | 8.5%     | 0.3 s  | 18.0%        |
> | CoT           | 0.5 s    | 10.0%    | 0.8 s  | 17.3%        |
> | CoT-SC (k=5)  | 2.3 s    | 11.0%    | 4.1 s  | 20.6%        |
> | CoT-SC (k=10) | 4.5 s    | 11.5%    | 8.8 s  | 20.6%        |
> | ToT           | -        | -        | 9.0 s  | 22.6%        |
> | VAP           | 4.1 s    | 16.5%    | 9.5 s  | 35.5%        |
>
> While VAP is indeed time-consuming, it achieves comparable time usage to ToT and CoT-SC (k=10) in the Sudoku task. The reason is that, although VAP's context is quite large (tool instructions, thought trajectory, image encoding), its output is concise (API calls and immediate thoughts), leading to fast inference during each step (time usage is mainly affected by the number of decoded tokens). We believe the time usage is acceptable given VAP's superior effectiveness.
>
> ### W1.2:Trade-off between the number of images and model performance
>
> This is an inspiring comment. Currently, the number of images is determined by the planner, and we don't provide access to modify it. To address the reviewer's concern, we add an experiment of controlling the number of images. The results of this experiment can be found in our response to all reviewers.
>
> ### W2/Q1:More LLM models need to be added for comprehensive comparison
>
> We follow the reviewer's advice and add GPT4 and LLaMA 3 8B as additional LLMs for more fairer comparison in our revision. The results can be found in our response to all reviewers. Based on results, we believe that the superiority of VAP over other baselines remains valid, even when utilizing different LLMs.
>
> ### W3:Effect of different figure drawing tools
>
> We sincerely thank your comment and it actually pushes us to think deeply about the relationship between tool selection and performance.
>
> First, we analyze the distribution of drawing tools selection:
>
> |            | Geometry | Sudoku | Time Series | TSP  |
> | ---------- | -------- | ------ | ----------- | ---- |
> | Matplotlib | 86.0%    | 91.3%  | 100%        | 78%  |
> | Turtle     | 14.0%    | 8.7%   | 0.0%        | 22%  |
> | DALLE3     | 0.0%     | 0.0%   | 0.0%        | 0.0% |
>
> Matplotlib was used more frequently, especially for time series prediction, due to its ability to construct coordinate systems efficiently.  DALLE3 is never selected by planner as the images of four tasks are all diagrams (we have explored another use case of DALLE3, detailed in Q2).
>
> Second, we are also interested in which tool perform better for these tasks. We conduct an experiment where we forced VAP to choose a specific tool. The results are as follows:
>
> |                 | Geometry↑ | Sudoku↑   | Time Series↓ | TSP(N=10)↓ |
> | --------------- | --------- | --------- | ------------ | ---------- |
> | Original        | 16.5%     | 35.5%     | 556          | 312.4      |
> | Matplotlib only | 16.5%     | **36.6%** | **556**      | 312.6      |
> | Turtle only     | 16.5%     | 19.0%     | 611          | **312.1**  |
>
> We find that the impact of different tools depends on the task. For geometry and TSP, using Matplotlib or Turtle makes no significant difference, while in Sudoku and time series forecasting, Matplotlib is better than Turtle. Because drawing grids and coordinates are too complicated for Turtle (draws by pen movement). The results also reflect the rationale behind the LLM's tool selection, as it automatically avoids choosing tools that are not suitable for a given task.
>
> ### W4: Scalability of VAP
>
> Solving complex reasoning problems has been a well-known weakness of LLMs and this domain has attracted significant attention because once the challanges can be overcome, many applications can well benefit from the power of LLMs. So, it is not surprising to find that the accuracy drops when facing challenging reasoning problems. Compared with other LLM-based reasoning frameworks, our VAP still demonstrates the best performance. The performance gap is even widened when the problem becomes more challenging.
>
> ### W5: Although VAP performs better than the baselines, its overall performance is still poor and has a noticeable gap compared to traditional methods
>
> We totally agree with the reviewer that the current performance of LLMs is still not comparable to traditional methods in many tasks. Its strength lies in generality, with one model to support vairious decision and prediction problems. Thus, how to exploit the potential of LLMs and improve its capabilities in reasoning and prediction has become a hot research topic in recent years. Our work falls in this category and we believe with more efforts devoted in this domain, the performance gap may continue to shrink.
>
> ### Q2: The potential of VAP in solving real-world reasoning problems, where the model generates more photorealistic images
>
> We've actually explored VAP's potential in a real-world sotrytelling task. The task setting is to write a story based on the input of text prompts.
>
> In the implementation of VAP, DALLE3 is always selected as the external image tool. Then at each iteration, VAP generates a photorealistic image to render a scene. This image is then sent to GPT-4v to enhance the model's creativity in story generation.
>
> Our previous findings show that, based on our subjective judgement, VAP can significantly benefit the task. We did not put this task in the manuscript due to the lack of a convincing performance metric.

---

> > ### Comment · Reviewer_gmDj · 2024-08-11
> >
> > Thank you for the author's detailed response. While most of my concerns have been addressed, I still have reservations regarding the scalability and generalizability of the VAP to more complex tasks. Since this is not my area of expertise, I recognize that VAP might represent an important step if it is indeed the first to achieve visual CoT by explicitly generating intermediate visual results. This could potentially influence my final rating, which I will determine during the reviewer discussion session. Otherwise, I encourage the author to provide more details about the current status of visual CoT.

---

> > > ### Author Response · Authors · 2024-08-14
> > > **Thank you for your feedback!**
> > >
> > > Thank you for your feedback and we understand your reservations about the scalability of VAP to complex tasks.
> > >
> > > First, we notice that solving complex reasoning problems is a well-known challenging for current LLMs-based methods. In our experiments, the compared LLM-based methods failed to solve the hard version of geometry intersection counting problems with shape = 6.
> > >
> > > | Method      | CoT  | CoT-SC(k = 5) | CoT-SC(k = 10) | CoT-SC(k = 20) | VAP      |
> > > | ----------- | ---- | ------------- | -------------- | -------------- | -------- |
> > > | Performance | 0.0% | 0.0%          | 0.0%           | 0.0%           | **7.5%** |
> > >
> > > While VAP's performance of 7.5% may seem low, all other baselines fail to solve any of these problems (with success rate 0%). It is noteworthy that these instances are quite challenging for humans as well. In this context, VAP's performance represents an improvement over existing LLMs-based methods.
> > >
> > > Second, we follow the reviewer's advice and provide more details about the current status of visual CoT. Here is a comparison with recent related work:
> > >
> > > | Method     | training-free ? | Main problem                                                 | Use self-systhetic image ? | Potential to solve complex numerical problems ? |
> > > | ---------- | --------------- | ------------------------------------------------------------ | -------------------------- | ----------------------------------------------- |
> > > | MMCoT      | 𐄂               | Visual question answer                                       | 𐄂                          | 𐄂                                               |
> > > | DDCoT      | 𐄂               | Visual question answer                                       | *𐄂*                        | 𐄂                                               |
> > > | Cantor     | *✓*             | Visual question answer                                       | 𐄂                          | 𐄂                                               |
> > > | ViperGPT   | *✓*             | Visual question answer                                       | 𐄂                          | *✓*                                             |
> > > | VisProg    | *✓*             | Visual question answer                                       | 𐄂                          | *✓*                                             |
> > > | LLava-Plus | 𐄂               | Visual question answer                                       | 𐄂                          | 𐄂                                               |
> > > | CoI        | 𐄂               | Numerical Reasoning problem (Geometry, chess)                | *✓*                        | *✓*                                             |
> > > | Visual CoT | *✓*             | stotytelling, summarization                                  | *✓*                        | 𐄂                                               |
> > > | VAP        | *✓*             | Numerical reasoning problems (Geometry, Sudoku, time series, TSP) | *✓*                        | *✓*                                             |
> > >
> > > While many works (e.g. MMCoT, ViperGPT) aim to use visual CoT for visual question answering, the input for those tasks consists of an image and a text question. **So** **these VQA-oriented approaches** **are unable to handle text-only problems by self-synthesizing images.** Another work, Visual CoT, uses generated photos to enrich the context for handling creative tasks such as storytelling and summarization, but it is not designed to solve complex numerical problems. CoI, on the other hand, is not a training-free method and requires task-specific training to learn the patterns of each task, making it unsuitable as a general reasoning method.
> > >
> > > We hope this additional context could help address your concerns.

---

### Official Review · Reviewer_vhJU · 2024-07-12

**Soundness:** 4
**Presentation:** 4
**Contribution:** 4
**Rating:** 8
**Confidence:** 5

**Summary:**

This paper targeting an intresting topi in VL research: Can VLM understand the organized prompts as LLM? The authors proposed a method called VAP(visual augmented prompt) to improve the prompting learning methods for VLM. The authors argue that human have two specialized subsystems that process verbal inforation and visual-spatial information respectively. Thus, to mimic human's decision making capability, the proposed method will synthesize images and organized a chain-of-thought in both modalities. This method comprises three steps: 1. selecting an anppropriate drawing toolkit and creating an initial image; 2. Iteratively perform reasoning on the synthesized images, and generates a paragraph of accompaning text for the generated image; 3. Finally, feed all the intermediate thoughts and images to the model as input to formulate the COT process.
This method is tested on 4 different tasks to show its general capability.

**Strengths:**

1. Though COT and prompt learning is not a novel topic in LLM. How to organize and build a CoT process for VLM to effectively activate the reasoning skill for VLM. While most recent work focusing on leveraging the language side to generate the rationale process. This work focusing on how to coordinately generate both visual and language rationale. This is a quite good idea.
2. The proposed method reflect the human decision process and thus sounds quite reasonable and novel.
3. The proposed method increase the results on different tasks by a large margin, even compare to many advance prompting and CoT styles. These solid increment prove the effectiveness of this method.

**Weaknesses:**

1. As mentioned by the authors. The image generation process not always controllable and could generate error-pone content to mislead the thinking process. Therefore I wonder, is generating a explicit image the best practice for this process. As the generation of images could result in unwanted features, we could also keep the representation in the latent space as a prompt.
2. The method relies on a planner to plan what to draw for the whole process. However, a planner itself can be incorrect and we simply have no control on this planner and the generated plan. Is there a more formulated way to generate the plan, or do we have somewhat of tools to detect the possible problem?

**Questions:**

1. Since the drawing steps and numbers are decided by the planner. What if the plan is quite long, and we are out of tokens? Do we have a control on the granularity on the intermediate drawing?
2. How frequent the generated images will content incorrect information? And how well the self-alignment module detect some very specific errors?

---

> ### Author Rebuttal · Authors · 2024-08-06
>
> We sincerely appreciate the constructive comments from the reviewer! We provide detailed responses to each concern in the following.
>
> ### W1: "I wonder, is generating an explicit image the best practice for this process. As the generation of images could result in unwanted features, we could also keep the representation in the latent space as a prompt."
>
> This is a really inspiring idea! In the current stage, commercial VLMs we used in this paper have not provided the API interface that allows us to use encoded image feature in the latent space as part of the input. We consider it as our future work to examine whether we can implement the idea on other open-sourced VLMs.
>
> ### W2: "Is there a more formulated way to generate the plan, or do we have somewhat of tools to detect the possible problem?"
>
> We appreciate your constructive feedback. We totally agree that a more formalized approach to generate the plan would likely improve the performance and robustness of our method.
>
> The ideas we currently have include:
>
> - Format control: we can utilize the open-source repository lm-format-enforcer (https://github.com/noamgat/lm-format-enforcer) to strictly formulate the output JSON schema. This tool uses a prefix tree decoder during the streaming output of the language model to enforce the output format.
> - Reflection mechanism: recent works, such as Reflexion, have shown that LLMs can self-reflect on previously generated answers and produce more reasonable responses. This mechanism can be applied on the planning stage, which could help detect potential problems.
> - Controllable parameters in planning prompt: we can introduce controllable parameters in the prompt of planning step, which would provide users with more control over the generated plan and would also contribute to performance (also proved in the experiment in Q1.2).
> - Planning decomposition: for complex tasks, decomposition can be an effective strategy. Techniques like ToT or ReAct could be applied to generate a more reliable plan.
>
> ### Q1.1: "Since the drawing steps and numbers are decided by the planner. What if the plan is quite long, and we are out of tokens?"
>
> Token length is indeed a common issue in LLM-based reasoning tasks. So far, the maximum number of tokens required by the planner of VAP is 2841, which is safely below the 8k token context limit in the default GPT-4 setting.
>
> ### Q1.2: "Do we have a control on the granularity on the intermediate drawing?"
>
> Thank you for your insightful comment. Currently, the granularity on drawing is determined by planner and we don't provide access to modify it. Inspired by your feedback, we explored the possibility of controlling granularity to enhance drawing efficiency.
>
> We conducted an experiment on Sudoku task by controlling the number of iterative steps (which is equivalent to controlling the granularity of drawing). Specifically, we inject the prompt "You must finish within `n_iterations` by drawing multiple rows in parallel" into the prompt of  iterative reasoning.  `n_iterations` is set to 8, 4, 2, and 1 to investigate performance. The results are as follows:
>
> |                        | Time usage | Correct rate |
> | ---------------------- | ---------- | ------------ |
> | VAP (original)         | 19.0 s     | 35.5%        |
> | VAP (n_iterations = 8) | 17.7 s     | 35.3%        |
> | VAP (n_iterations = 4) | 15.9 s     | **37.3%**    |
> | VAP (n_iterations = 2) | 12.3 s     | 26.6%        |
> | VAP (n_iterations = 1) | 4.4 s      | 19.9%        |
>
> It is interesting to find that set `n_iterations` to 4 improve performance over the original version and also enhancing efficiency. This suggests that more iterations do not necessarily lead to higher accuracy. When `n_iterations` is set to 1, the iterative reasoning process is almost skipped, resulting in poor performance. These findings support your previous comment (W2) that a more formulated generation approach is beneficial and worth further exploration. We appreciate the reviewer's comment for pointing out this.
>
> ### Q2: "How frequent the generated images will content incorrect information? And how well the self-alignment module detect some very specific errors?"
>
> We analyzed the frequency of generated images containing incorrect information and report the results in the following table:
>
> |             | Correct rate without self-alignment | Correct rate with self-alignment |
> | ----------- | ----------------------------------- | -------------------------------- |
> | Geometry    | 77.0%                               | 83.5%                            |
> | Sudoku      | 72.0%                               | 89.3%                            |
> | Time Series | 100.0%                              | 100.0%                           |
> | TSP         | 98.0%                               | 98.0%                            |
>
> We define an image as "correct" if it contains the integral information described in the text. From the result, we observe that the geometry and Sudoku benefited significantly from self-alignment. In geometry task, LLM would occasionally initialize coordinates with the wrong range, causing the shapes outside of the view. In Sudoku, LLM may have difficulty translating position descriptions to image coordinates. For example, in a 9x9 board, the correct API call grid position "1 2 4" should be translated to `plt.text("1", x=1.5, y=3.5)`(placing the element in the center of the grid). However, the LLM occasionally make mistakes, such as:
>
> - `plt.text("1", x=2, y=4)`, placing at the bottom right corner
> - `plt.text("1", x=1, y=3)`, placing at the top left corner
> - Missing this element
>
> These mistakes can be detected by self-alignment because the LLM will describe the content of the image and check it against the original input, which is proved to be crucial in our ablation study.

---

> > ### Comment · Reviewer_vhJU · 2024-08-07
> >
> > Thank you for the authors response. I find this response very satisfying as they include new experiments results as I required. Since I give an "8" as my initial rating, which is a very high score, I would not change it anymore. But I still recommend this paper to be accepted and enrouge other reviewers to raise their ratings as this paper really addressed some under studied topic of Vision-Language models. I wish the authors could include the discussion in this response into their main content later to make this work more complete. Wish you good luck.

---

### Official Review · Reviewer_L2VS · 2024-07-12

**Soundness:** 3
**Presentation:** 3
**Contribution:** 3
**Rating:** 5
**Confidence:** 4

**Summary:**

This paper proposes a new prompting technique, vision-augmented prompting (VAP), to improve the reasoning capabilities of large language models (LLMs). Different from the mainstream chain-of-thought (CoT) frameworks that only involve textual reasoning steps, the proposed VAP framework automatically generates images from visual and spatial clues via external tools. In addition, the VAP framework feeds a chain of thought with both textual and visual context into LLMs to solve the original problems. Evaluations on four tasks (i.e., geometry, sudoku, time series prediction, travelling salesman problem) demonstrate that the proposed VAP outperforms prior CoT frameworks.

**Strengths:**

1. The proposed VAP framework improves the traditional chain-of-thought (CoT) prompting via augmenting visual context. Specifically, the VAP generates high-level drawing planning from the textual question, and then iteratively outputs instructions to draw the stepwise visual context and provide related textual thoughts. The visual contexts with textual context (i.e., thoughts) provide richer information than the text-only context in the CoT framework and helps the reasoning capability of the LLM (MLLM) models.

2. As the iteratively generated visual and textual contexts may contain errors and lead to wrong reasoning, the authors propose a self-alignment mechanism to check if the visual and textual contexts align with the initial high-level drawing planning. If not aligned, the iterative reasoning procedure will be restarted accordingly. This mechanism helps to improve the iterative reasoning and results in a more accurate conclusion.

3. In the evaluation section, the authors conduct a comprehensive comparison between the proposed models and multiple CoT frameworks on 4 versatile benchmarks. In addition, the authors also introduce several task-specific baselines during the comparison and it further validates the effectiveness of the proposed VAP framework. Moreover, the authors conduct human analysis on some generated drawing and observe an impressive integrity rate. These evaluations are helpful to understanding the strengths of the VAP framework.

4. The paper is well written and easy to read.

**Weaknesses:**

1. In the VAP framework, it chooses one of three tools (matplotlib, turtle, and dalle3), but the manuscript does not provide much details about how these tools are used
- For matplotlib and turtle, code generation is needed. However, how can the framework guarantee that the code has the right syntax to properly generate the image?
- For dalle3, how is the prompt generated for the image drawing?
- What is the distribution of calls on these three different tools?
- In each iteration of iteratively reasoning, does the selected tool draw a complete image, or overlay the new drawing on image from previous iteration?
- Can the framework use a mixture of different tools in resolving one problem?

2. In the evaluation, the questions from all benchmarks are provided in text format. I wonder if the authors can include benchmarks with both text and image in question (e.g., VQA benchmarks). This is a fairer setting since the current baselines ignore the visual capability of the GPT4v model.

3. In ablation studies, there is one experiment that removes the planning step. I wonder if the authors can provide more details. Does it mean that a different set of iterative reasoning prompt is used?

4. Typos: Line 294: is plays -> plays / "Self-alignment plays an important role to the task of Geometry Intersection Problems" Shouldn't it be "Sudoku" since self-alignment achieves the largest improvement (25.1% -> 35.5%) on Sudoku?

**Questions:**

See the Weaknesses section for detailed questions.

**Limitations:**

The authors mentioned and attempted to address several limitations in the manuscript.

---

> ### Author Rebuttal · Authors · 2024-08-06
>
> We thank the reviewer for the in-depth review! Below, we respond to the weaknesses raised in the review.
> ### W1.1:"For matplotlib and turtle, how can the framework guarantee the syntax correctness of generated code?"
> We understand the reviewer's concern. We add an experiment for code generation assessment and find the LLM-based code generation is sufficiently robust to guarantee syntax correctness by itself. The following table presents the ratio of syntax error and runtime error:
> |             | Syntax Error | Runtime Error |
> | ----------- | ------------ | ------------- |
> | Geometry    | 0%           | 0%            |
> | Sudoku      | 0%           | 2.7%          |
> | Time Series | 0%           | 0%            |
> | TSP         | 0%           | 0%            |
>
> Here, LLM produces no syntax errors. Because the code generation for API call is not challenging for LLM, without involving complex logic. For example,  "a circle center at (1,3) with radius 2" is translated to `plt.circle(c=(1,3),r=2)`. We can observe slight runtime error in the Sudoku task. These errors occurred when translating Sudoku positions to actual coordinates in the image. For example, in a 9x9 board, the grid position "1 9 1" should be translated to `plt.text("1",x=0.5,y=8.5)`(placed in the center of the grid). However, the LLM occasionally made mistakes like `plt.text("1",x=1,y=9)`, causing out-of-bounds placement.
> ### W1.2: "For dalle3, how is the prompt generated for the image drawing?"
> For DALLE3, LLM also generates an API call to create the image(OpenAI offers a package for DALLE3).
>
> As DALLE3 is not selected in the tasks presented in our experiment(detailed in W1.3), we take another task we have previously tried as the example. This task is creative writing task, where LLM is asked to write storied based on some keywords.(We did't put the task in manuscript due to the lack of a convincing performance metric)
>
> Example of this task:
>
> Input: "Write a story according to given key words: Mage, Warriors, Priest"
>
> Output: "A light breeze swept the ground, ..."
>
> In this example, VAP first make a plan like:
> ```json
> {
>   "tool": "DALLE3",
>   "initialization": "dalle3 = OpenAI().images",
>   ...
> }
> ```
> Each iteration, VAP will generate an image using API call. For example, `img = dalle3.generate(prompt="The Mage dressed in a dark cloak, holding a staff and surrounded by a magical aura.", size="1024x1024")`.
> ### W1.3: "What is the distribution of calls on these three different tools?"
> As responded in W1.2, DALLE3 is not selected in the four tasks. So, we report the distribution of calls on the tools of Matplotlib and Turtle in the following:
> |            | Geometry | Sudoku | Time Series | TSP  |
> | ---------- | -------- | ------ | ----------- | ---- |
> | Matplotlib | 86.0%    | 91.3%  | 100%        | 78%  |
> | Turtle     | 14.0%    | 8.7%   | 0.0%        | 22%  |
>
> Matplotlib is used more frequently, especially for time series prediction, due to its ability to construct coordinate systems efficiently.
>
> We've also explored the relation between selected tool and performance, which can be found in response to Reviewer gmDj.
> ### W1.4: "In iterative reasoning, the selected tool draw a complete image, or overlay the new drawings from previous iteration?"
> In each iteration, our method uses the drawing tool to overlay the new drawing on the image from the previous iterations.
>
> For example, consider a geometry problem with two shapes, as follows:
> ```Plain
> There's a circle centered at (3, 2) with radius 1
> There's a circle centered at (6, 1) with radius 4
> How many intersection points are there?
> ```
> In the second iteration, the LLM will use the API call `plt.circle(c=(6,1),r=4)` to draw an additional circle on top of the previous image incrementally.
> ### W1.5: "Can the framework use a mixture of different tools in resolving one problem?"
> Unfortunately, our method does not support using a mixture of different tools during iterative reasoning. The drawing tool is determined in the planning step by LLM and remains fixed to ensure consistency.
> ### W2: "I wonder if the authors can include benchmarks with both text and image in question (e.g., VQA benchmarks). This is a fairer setting since the current baselines ignore the visual capability of the GPT4v model."
> We understand the reviewer's concern regarding the fairness of baseline comparision. It can be challenging to create an absolutely fair setting when baselines only require text ability of VLLM. Our ablation study may provides valuable insights. When the iterative reasoning stage is removed, it downgrade VAP to standard prompting with visual ability. And the results show a clear superiority of VAP over such a baseline.
>
> We also take the reviewer’s advice to investigate VQA benchmarks. However, we find our task setting differs from VQA. Our work focuses on enhancing text-only problems using the additional image channel, while VQA typically involves answering questions based on provided images. This represents a distinct area of research.
> ### W3: "In ablation studies, there is one experiment that removes the planning step. I wonder if the authors can provide more details. Does it mean that a different set of iterative reasoning prompt is used?"
> We apologize for the lack of details. When the planning step is removed, we lose access to meta-information like selected tool, which is used to fill into iterative reasoning prompt template. Therefore, we use an alternative prompt template for iterative reasoning. The key changes to this new prompt will be detailed in the Appendix in revision. The key changes of this prompt include:
> - No specific draw content and thought content given
> > ..., provide your thoughts according to this problem.
> - No specific tool given
> > ... update the image content using Python API calls.
> ### W4: "Typos at Line 294"
> Thank you for pointing these out. We have fixed the typo and it's indeed that Sudoku benefits most from self-alignment. We have revised the statements accordingly.

---

> > ### Comment · Reviewer_L2VS · 2024-08-14
> > **Thank you for the rebuttal**
> >
> > Thank you for the detailed rebuttal. I think most of my concerns have been addressed.

---

### Official Review · Reviewer_VnVX · 2024-07-13

**Soundness:** 3
**Presentation:** 3
**Contribution:** 3
**Rating:** 5
**Confidence:** 3

**Summary:**

This paper proposes visual-augmented prompting (VAP) for large language models (LLMs) in reasoning tasks. Specifically, VAP translates textual questions into a sequence of self-synthesized images using API calls (Python Turtle, Matplotlib, DALL-E3). These images are then fed back to Vision-LLM (GPT-4o) in a step-by-step manner as deduction steps. The detailed VAP process includes (1) planning, (2) iterative reasoning, and (3) conclusive reasoning. Experiments on several math tasks, such as geometry intersection counting, Sudoku puzzles, time series prediction, and the traveling salesperson problem, demonstrate that VAP helps LLMs perform better than chain-of-thought (CoT) and tree-of-thought (ToT) methods.

**Strengths:**

1. The paper extends Chain-of-Thought (CoT) with visual prompt information. In addition to the step-by-step textual deduction in CoT, it uses drawing APIs (e.g., Python Turtle, Matplotlib, DALL-E) to synthesize pictures in the intermediate steps, helping to derive the final answer for mathematical problems. This approach is interesting and has not been explored before.

2. Experiments on math-related problems, such as geometry intersection counting, Sudoku puzzles, time series prediction, and the traveling salesperson problem, demonstrate that VAP outperforms CoT and ToT methods.

**Weaknesses:**

1. When the LLM generates Python API calls, there is a chance that the generated code may not run successfully (bugs in the code). What is the probability of this phenomenon occurring in the experiments?

2. Generalization: CoT and ToT are more generalized to different LLM tasks, while VAP is limited to a few geometry problems. Does it generalize to normal Visual QA tasks?

3. There is a missing reference to relevant works on tool usage ability for LLMs, such as ViperGPT [1], VisProg [2], and LLava-Plus[3].


[1]. ViperGPT: Visual Inference via Python Execution for Reasoning
[2].Visual Programming for Compositional Visual Reasoning
[3]. LLaVA-Plus: Learning to Use Tools for Creating Multimodal Agents

**Questions:**

1. What is the chance of visual drawing code failing? How to reduce this probability.

2. Does the model require customized instructional prompts for LLM on different tasks?

3. Can the model generalize to more general Visual QA tasks?

4. Is it possible to compare with VCoT mentioned in Section 2.3 (line 76)?

**Limitations:**

The authors discuss and address some of the limitations of VAP.

---

> ### Author Rebuttal · Authors · 2024-08-06
>
> We sincerely appreciate the constructive comments from the reviewer. We provide detailed responses to each concern in the following.
>
> ### W1&Q1: "When the LLM generates Python API calls, there is a chance that the generated code may not run successfully (bugs in the code). What is the probability of this phenomenon occurring in the experiments?"
>
> We understand the reviewer's concern and have added an experiment to assess the code generation error rate. The following table presents the ratio of syntax error and runtime error:
>
> |             | Syntax Error Rate | Runtime Error Rate |
> | ----------- | ----------------- | ------------------ |
> | Geometry    | 0%                | 0%                 |
> | Sudoku      | 0%                | 2.7%               |
> | Time Series | 0%                | 0%                 |
> | TSP         | 0%                | 0%                 |
>
> The results show that the componnet of LLM-based code generation is sufficiently robust, with no syntax error. This is because the task of code generation for API call is not challenging for LLM, without involving complex logic. For example,  "a circle center at (1, -1) with radius 2" is translated to `plt.circle(c=(1, -1), r=2)`. We can observe slight runtime error in the Sudoku task. These errors occurred when translating Sudoku positions to actual coordinates in the image. For example, in a 9x9 board, the grid position "1 9 1" should be translated to `plt.text("1", x=0.5, y=8.5)`(placing the text in the center of the grid). However, the LLM occasionally made mistakes like `plt.text("1", x=1, y=9)`, causing out-of-bounds placement.
>
> ### W2&Q3: "Generalization: CoT and ToT are more generalized to different LLM tasks, while VAP is limited to a few geometry problems. Does it generalize to normal Visual QA tasks?"
>
> Yes, we agree with the reviewer that CoT and ToT are more generalized, but their performances are inferior to our VAP in the four diversified reasoning tasks that can benefit from dual-modality reasoning.
>
> In the current stage, we are unable to support normal visual QA tasks for two main reasons. First, the problem setting is different. The input of visual QA tasks consists of an image and text question. In our setting, the input is a text question. The image is automatically synthesized according to the text input. Second, VAP is designed to improve numeric reasoning, whereas normal visual QA tasks focus on semantic understanding of the input image. Nonetheless, we agree that it would be very impactful if we can extend VAP to be more general to support VQA.
>
> ### W3: "There is a missing reference to relevant works on tool usage ability for LLMs, such as ViperGPT, VisProg, and LLava-Plus."
>
> We thank the reviewer for providing these relevant works. In revision, we incorporated these works in related work and add necessary explanations and discussions.
>
> ### Q2: "Does the model require customized instructional prompts for LLM on different tasks?"
>
> In fact, VAP does not require task-specific prompts for different tasks, which we think is a desirable feature. This ows to the planning step, in which the prompt specifies the role, drawing workflow, external image tools, and output format. These elements are identical for different tasks. The prompt also contains the input problem text, which triggers LLM to understand the task and generate different plans accordingly.
>
> The key components of the prompt template for the planning include:
>
> > Your role is to visualize a problem by creating an image... // Role play
> >
> > The drawing will be executed through an iterative process... // Workflow clarification
> >
> > Here are drawing tools available:... // Introduction of the drawing tools
> >
> > You must strictly follow the JSON format... // Format control
> >
> > Problem Description:... // Problem input
>
> More details on this prompt can be found in Appendix A.1.1.
>
> ### Q4: "Is it possible to compare with VCoT mentioned in Section 2.3 (line 76)?"
>
> VCoT was designed for tasks like storytelling and summarization, instead of the numeric reasoning tasks presented in our work. Furthermore, we found that its source code is not available, with an empty page in the provided github link. To address the reviewer's concern, we came up with an idea to apply our VAP to the sotrytelling task as presented in the VCoT paper. The task setting is to write a story based on the input of text prompts.
>
> In the implementation of VAP, we incorporate DALLE3 as one of the external image tools. Then at each iterative step, VAP generates a photorealistic image using DALLE3 to render a scene. This image is then sent to GPT-4v to enhance the model's creativity in story generation.
>
> As to performace metric, VCoT adopts human evaluation to judge creativity etc, which we think are subjective. Therefore, for reviewer's reference, we simply provided an example below to demonstrate the results of two approaches.
>
> | Input short prompt                          | Me and my son decided to take a nice walk on the beach.We both were admiring the scenery. Afterwards, we started driving home. |
> | ------------------------------------------- | ------------------------------------------------------------ |
> | Result of VCoT (reported in original paper) | On the way home from school, my son and I enjoyed the beautiful view of the sky and flowers.We had picked up a presentation of flowers from school, which we admired as we drove home.The sky was blue and white over the car, and we enjoyed the scenic view of the flowers and the sky. |
> | Result of VAP                               | The sun dipped below the horizon as my son and I strolled along the sandy shore, admiring the beautiful colors of the sky and the calming sound of the waves.We were both quiet, watching the seagulls fly above us, the interesting shapes in the sand, and the boats far out on the water.As the day ended, we walked back to our car, feeling happy and closer than ever after our special time together on the beach. |

---

### Author Rebuttal · Authors · 2024-08-06

We thank all detailed feedbacks provided by the reviewers! We address a few points in this response. All other questions are addressed in reviewer specific responses.



### Trade-off between number of iterations and performance

We add an experiment of controlling the number of iterations in Sudoku task. Specifically, we injected the prompt "You must finish within `n_iterations` by drawing multiple rows in parallel" into the prompt of iterative reasoning. `n_iterations` is set to 8,4,2 and 1. The results are as follows:

|                    | Time   | Correct rate |
| ------------------ | ------ | ------------ |
| VAP (original)     | 19.0 s | 35.5%        |
| VAP (iterations=8) | 17.7 s | 35.3%        |
| VAP (iterations=4) | 15.9 s | **37.3%**    |
| VAP (iterations=2) | 12.3 s | 26.6%        |
| VAP (iterations=1) | 4.4 s  | 19.9%        |

It is interesting to find that set `n_iterations` to 4 improve performance over the original version and also enhancing efficiency. This suggests that more iterations do not necessarily lead to higher accuracy. When `n_iterations` is set to 1, the iterative reasoning process is almost skipped, resulting in poor performance.



### Baselines comparasion with different LLMs

Our experimental setup employs a unified VLLM. However, we observed that VAP necessitates a MLLM to process visual-text input, whereas other baselines solely require textual input. Considering that traditional LLMs are expected to outperform MLLMs in text-only tasks, we introduced GPT-4 and LLaMA 3 8B as additional LLMs to ensure a more comprehensive comparison. Accuracy on the geometry task is presented as follows:

|               | GPT-4v    | GPT4  | LLaMA3 |
| ------------- | --------- | ----- | ------ |
| Standard      | 8.5%      | 10.0% | 7.0%   |
| CoT           | 10.0%     | 11.0% | 8.0%   |
| CoT-SC (k=5)  | 11.0%     | 11.5% | 8.0%   |
| CoT-SC (k=10) | 11.5%     | 11.5% | 8.0%   |
| CoT-SC (k=20) | 11.5%     | 11.5% | 8.0%   |
| VAP           | **16.5%** | -     | -      |

Here, we can get these conclusions:

- GPT-4 slightly improved baseline performance compared to GPT-4v, but still significantly lower than VAP.
- LLaMA3 8B shows decreased accuracy, likely due to its small size limiting generalizability. Note that currently we are unable to run larger versions of LLaMA3 due to machine constrain.
- Simpler methods (standard prompting) benefit more from model changes than complex methods (CoT-SC).

 From the results, we maintain that the superiority of VAP over other baselines remains valid.

---

### Decision · Program_Chairs · 2024-09-25

**Decision:**

Accept (spotlight)

**Comment:**

This paper proposes visual-augmented prompting (VAP) framework that automatically translates textual questions into a sequence of self-synthesized images from visual and spatial clues using external tools with API calls (Python Turtle, Matplotlib, DALL-E3). These images are then fed back to Vision-LLM (GPT-4o) in a step-by-step manner as deduction steps. The detailed VAP process includes (1) planning, (2) iterative reasoning, and (3) conclusive reasoning. Experiments on several math tasks, such as geometry intersection counting, Sudoku puzzles, time series prediction, and the traveling salesperson problem, demonstrate that VAP helps LLMs perform better than chain-of-thought (CoT) and tree-of-thought (ToT) methods.

In general image generation process is not always controllable and could generate error-pone content to mislead the thinking process. During rebuttal, the authors analyzed the frequency of generated images containing incorrect information. From the result, it is observed that the geometry and Sudoku benefited significantly from self-alignment. In geometry task, LLM would occasionally initialize coordinates with the wrong range, causing the shapes outside of the view. In Sudoku, LLM may have difficulty translating position descriptions to image coordinates